# Building extraction from remote sensing imagery using SegFormer with post-processing optimization

Deliang Li[1,2,3,4], Tao Liu[1,2,3], Haokun Wang[5*], Long Yan[6]

1 Faculty of Geomatics, Lanzhou Jiaotong University, Lanzhou, Gansu, China, 2 National-Local Joint Engineering Research Center of Technologies and Applications for National Geographic State Monitoring, Lanzhou, Gansu, China, 3 Key Laboratory of Science and Technology in Surveying & Mapping, Lanzhou, Gansu Province, China, 4 Songyuan Vocational and Technical College, Songyuan, China, 5 Baicheng Normal University, Baicheng, China, 6 Middle School of Jilin City, Jilin, China

* wanghaokun@bcnu.edu.cn

## Abstract

Traditional methods for building extraction from remote sensing images rely on feature classification techniques, which often suffer from high usage thresholds, cumbersome data processing, slow recognition speeds, and poor adaptability. With the rapid advancement of artificial intelligence, particularly machine learning and deep learning, significant progress has been achieved in the intelligent extraction of remote sensing images. Building extraction plays a crucial role in geographic information applications, such as urban planning, resource management, and ecological protection. This study proposes an efficient and accurate building extraction method based on the SegFormer model, a state-of-the-art Transformer-based architecture for semantic segmentation. The workflow includes data preparation, model construction, model deployment, and application. The SegFormer model is selected for its hierarchical Transformer encoder and lightweight MLP decoder, which enable high-precision binary classification of buildings in remote sensing images. Additionally, post-processing techniques, such as noise filtering, boundary cleanup, and building regularization, are applied to refine the inference results, significantly improving both the visual presentation and accuracy of the extracted buildings. Experimental validation is conducted using the publicly available WHU building dataset, demonstrating the effectiveness of the proposed method in urban, rural, and mountainous areas. The results show that the SegFormer model achieves high accuracy, with the MiT-B5 backbone network reaching 94.13% Intersection over Union (IoU) after 100 training epochs. The study highlights the robustness and scalability of the method, providing a solid technical foundation for remote sensing image analysis and practical applications in geographic information systems.

**Data availability statement:** All minimal data and code necessary to replicate the findings reported in this study are openly available on the Figshare repository under the CC BY 4.0 license. The persistent identifier (DOI) for the dataset is: https://doi.org/10.6084/m9.figshare.c.8004457.

**Funding:** This research was supported by the Jilin Province Higher Education Reform Research Project "Research and Practice on Collaborative-Driven Innovative Talent Training Model for Geographic Information Science Majors" [grant no. 20224BR01C500HE]; the Ministry of Education Industry-University Cooperation Collaborative Education Project [grant no. 230902313194605]; and the Songyuan Vocational and Technical College Project "Research on VR-Integrated UAV Training Teaching Model" [grant no. SZ25013].

**Competing interests:** The authors have declared that no competing interests exist.

## Introduction

Buildings represent important feature targets that reflect geographic information in remote sensing images, making the extraction of buildings particularly significant for surface coverage classification, urban planning, disaster emergency assessment, and geographic information database updates. With the advancement of remote sensing technology and the widespread application of remote sensing image data, the development of efficient and accurate building extraction methods has emerged as a prominent research focus [1].

Traditional building extraction methods primarily rely on hand-designed features and algorithms such as threshold segmentation, texture features, and shape features [2]. However, these approaches are limited by low extraction accuracy and poor robustness due to the complexity and variability of remote sensing images. Consequently, research on deep learning based building extraction is considered highly significant. Recent years have witnessed remarkable success of deep learning in computer vision, demonstrating substantial potential for building extraction tasks. Notably, methods based on convolutional neural network [3] have proven effective in improving extraction accuracy and robustness through learning high-level image feature representations [4].

The present study aims to develop an efficient and accurate building extraction method for remote sensing images using the SegFormer model [5,6]. Specifically, data prepare will be conducted to generate training datasets. Subsequently, a deep learning model will be constructed based on SegFormer to achieve accurate building extraction through feature learning and model training, followed by post-processing of inference results. Finally, experimental validation will be performed to assess the method's performance and effectiveness.

The main contributions of this work include:

(1) The proposal of a novel building extraction method that addresses the low efficiency of traditional approaches;

(2) Optimization of building representation through post-processing of SegFormer-based extraction data.

The remainder of the paper is organized as follows: Section 2 reviews the related literature; Section 3 details the datasets and methodology; Section 4 presents the experimental results; Section 5 provides critical discussion; and Section 6 concludes the study.

## Related work

### SegFormer

SegFormer is a neural network architecture for semantic segmentation tasks, proposed in 2021 by Enze Xie et al [5]. Semantic segmentation is an important and challenging task in computer vision, where the goal is to segment an image into multiple regions, each of which corresponds to a specific category or object [3]. The design of SegFormer is inspired by previous successful semantic segmentation methods,

such as DeepLab [7] and Mask R-CNN [8]. Before SegFormer was proposed, there existed many other neural network architectures for image segmentation such as FCN [9], U-Net [10], Mask R-CNN, FPN [9], DeepLab, BiSeNet [11,12] and OCRNet [5,12]. These methods typically require large amounts of labeled data and training time [13]. However, the design goal of SegFormer is to realize a simpler and more efficient approach to achieve better performance in various scenarios.

In recent years, with the continuous development of deep learning technology, SegFormer has been widely used in the field of NLP and has received extensive attention and research. In the research of SegFormer, the main problem at present is how to improve the accuracy and efficiency of the model. In order to solve this problem, researchers have used various methods, such as improving the model structure, optimizing the training algorithm [14], and increasing the amount of data [15]. In addition, the interpretability of SegFormer is also one of the hotspots of research. By analyzing the internal mechanism and decision-making process of the model, researchers revealed the decision-making law and influencing factors of SegFormer, which provided ideas and methods to further improve the efficiency and accuracy of the model.

SegFormer stresses robustness and validity in Transformer series networks. While having challenges against image interference, the semantic segmentation task is completed efficiently and with higher precision. Issues of building detection in remote sensing images can be solved better due to the improvements made based on SegFormer. It has the following improvements in addition to the potential of high processing efficiency and dense prediction of Transformer series.

(1) A novel positional-encoding-free and hierarchical Transformer encoder.

(2) A lightweight decoder is constructed and MLP is issued for feature aggregation.

## Building extraction

Building extraction from remote sensing images is an important research direction in the field of remote sensing, which provides important data support for urban planning [16,17], Landslide [18], land use [19] and other related fields by recognizing and extracting buildings in remote sensing images. In recent years, with the continuous development and application of remote sensing technology, the research on building extraction from remote sensing images has also received more and more attention.

In the study of building extraction from remote sensing images, the following problems are mainly faced: first, the geometry of buildings is complex, and different buildings have different features, so how to accurately distinguish and recognize these buildings is a difficult problem [20]; second, the similarity between buildings and other objects is large, such as roads, trees, etc., so how to make an effective distinction is also a challenge [21,22]; third, in the remote sensing images, there are noise [12] and interference [23], how to remove these interferences and improve the accuracy of building extraction is also an important issue. In order to solve the above problems, researchers have used a variety of methods, such as machine learning-based methods [24], feature extraction-based methods [25–27], and deep learning-based methods [28]. Among them, machine learning based methods are widely used in remote sensing image building extraction, which automatically recognizes and extracts buildings by building models. The deep learning-based methods, on the other hand, pay more attention to feature extraction and classification of buildings, with high accuracy [29] and robustness [30].

## Building extraction post processing

Remote sensing image deep learning inference results can be optimized and refined by a series of data post-processing tools [31]. Among them, plurality filtering can be used to remove noise and artifacts [32]; region aggregation can merge multiple images into one, reducing the amount of data and improving the accuracy; boundary cleanup can remove boundary lines in the image to make the image smoother; refinement can further process the image to improve accuracy; building regularization [33] can process building region with rules such as right angle, rounded angle, arbitrary angle, diagonal, etc. to improve display effect.

## Methodology

Fig 1 shows the framework for building extraction. The public dataset WHU building dataset for building extraction. The overall workflow is as follows: (1) Data preparation: data preparation includes data collection, data preprocessing and training data generation. (2) Build extraction model: this paper uses SegFormer model. Use mit-b3 and mit-b5 backbone. (3) Model application: model inference and post-processing and model evaluation.

### Data preparation

Data preparation is usually the most important part of the entire deep learning process, and data quality largely determines model accuracy. Deep learning uses labeled data to train neural networks, which we call sample data. Sample data mainly consists of two parts: images and labels, and commonly used image data include satellite images, aerial photos, drone images, etc. Different regions have different cultural beliefs, folk customs, and different religious and legal systems [34], which produce different national ideologies, and these ideologies can also be reflected in buildings, which, as a carrier of local life, inevitably receive geographical influences [35]. Therefore, we extract the buildings, in order to provide the accuracy of deep learning extraction, try to be in the same geographical area where the building styles are similar. When partitioning image data into varying-sized training samples, the influence of feature extraction varies with the input dimensions. Larger image slices inherently retain more comprehensive feature representations, yet they impose greater computational demands on memory and processing resources. Consequently, the optimal slice size should be determined through a balanced consideration of the model architecture and the available hardware capabilities, ensuring efficient training without exceeding computational constraints. This parameter is also related to the spatial resolution of the image, for example, in order to extract the buildings, the width of the buildings in the 0.05 m spatial resolution image in hundreds of pixels or so, at this time can be appropriately increased image size to enhance the building information of a single

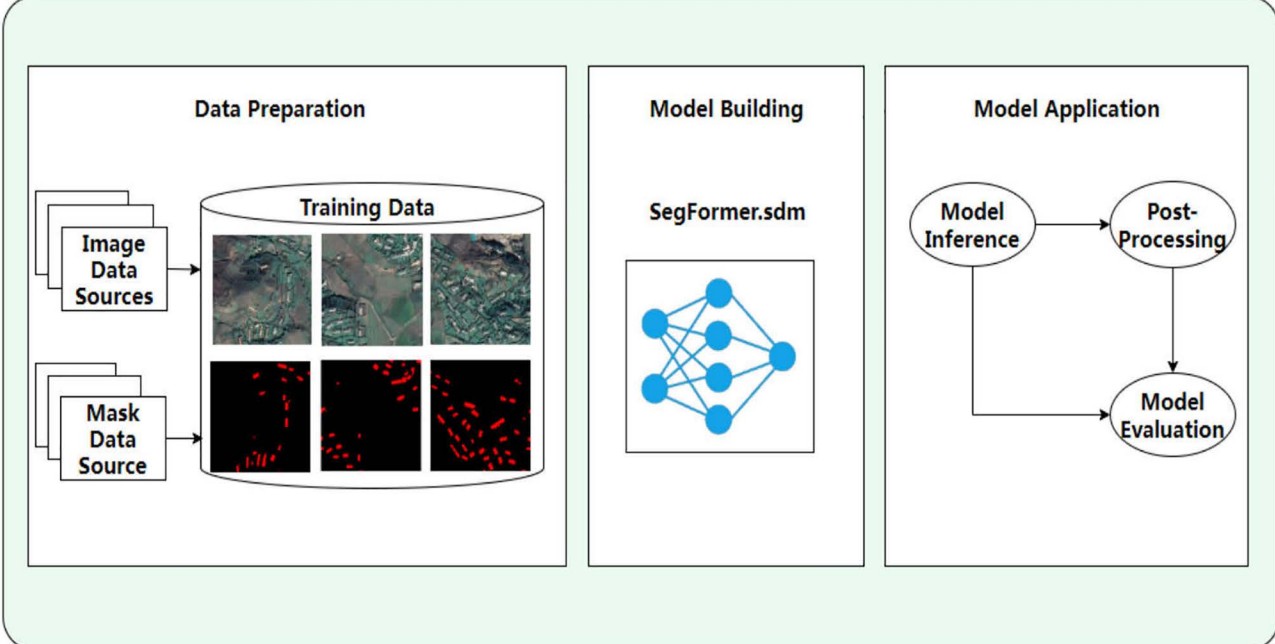

**Fig 1. A research framework for building extraction from remote sensing images.** This schematic diagram is redrawn based on experimental results and is not exactly the same as the original annotated images provided in the WHU dataset. It is only for illustrative purposes. https://doi.org/10.6084/m9.figshare.30041062.

picture; and in the 1 m spatial resolution image the width of the buildings may occupy dozens of pixels or even dozens of pixels, so the slice size can be set to be relatively small in order to reduce the proportion of training data of non-buildings, thus increasing the training data of non-buildings. Therefore, the slice size can be set relatively small to reduce the proportion of non-building training data and increase the proportion of building samples.

The binary classification training data includes image slices, sample labels, and training data metadata [36]. the Images folder is used to save the image slices; the sample labels are saved in the form of 0 and 1 binary rasters to the Masks folder, where 1 is the feature of interest and 0 is the background value; and the basic information of the training data is recorded in the training data metadata file.

## Model building

Model construction is mainly to let the neural network fully learn the feature information in the training data, and finally get the deep learning model. However, sometimes due to poor data quality and insufficient training, it can lead to problems such as insufficient model accuracy and poor generalization ability, so there is usually a model tuning step after the model effect verification.

Mainstream GIS products support a variety of modeling algorithms, such as SFNet, D-LinkNet, DeepLabV3+, Seg-Former and other binary classification algorithms, and each model has a corresponding applicable scenario. For example, D-LinkNet is suitable for line feature extraction; SFNet has fewer model parameters, which is convenient for training and suitable for model training with small data volume; DeepLab V3+ can better segment multi-class features; SegFormer adopts positionless encoder with Transformer technology, which is compatible with images of different resolutions. Users can select different modeling algorithms according to the task characteristics. In this paper, we adopt the SegFormer model, and the core of SegFormer is a hierarchical Transformer encoder with positionless coding and a lightweight all-MLP decoder, which can be easily adapted to arbitrary resolutions without affecting the performance. Whereas the layering of the encoder generates fine features for high resolution and coarse features for low resolution, the MLP decoder aggregates multiple layers of features to produce the final result. Overall, SegFormer is a simple, efficient and powerful modeling algorithm that simultaneously balances effectiveness, efficiency, and robustness.

In this paper, the SegFormer model is used for building extraction. SegFormer consists of two main modules: (1) a hierarchical Transformer encoder; and (2) a lightweight All-MLP decoder to predict the final mask. Given an image with size $H \times W \times 3$, we first divide it into patches of size $4 \times 4$. Unlike ViT which uses $16 \times 16$, using fine-grained patches favors semantic segmentation. Second, we use these patches as input to the hierarchical Transformer encoder to get multi-level features with resolution {1/4, 1/8, 1/16, 1/32} of the original image. We then pass these multi-level features to the All-MLP decoder to predict the segmentation mask with a $H/4 \times W/4 \times N_{cls}$ resolution, where $N_{cls}$ is the number of categories. The structure of the SegFormer network model is shown in Fig 2.

**(1) Encoder.** SegFormer has a series of Mix Transformer encoders (MiT), MiT-B0 to MiT-B5, with the same architecture but different sizes. MiT-B0 is the lightweight model we use for fast inference, while MiT-B5 is the largest model with the best performance. MiT-B3 is able to achieve high segmentation accuracy and recall while keeping low computational complexity. MiT-B5 is the best performing model. MiT-B3 and MiT-B5 are chosen for this experiment.

**(2) Decoder.** SegFormer integrates a lightweight decoder consisting of only MLP layers, which avoids the need for hand-crafted and computationally demanding components typically used in other approaches. The key to achieving this simple decoder is that our hierarchical Transformer encoder has a larger effective reception field (ERF) than traditional CNN encoders.

**(3) Other configurations.** In deep learning models, the parameters that need to be configured according to the situation before training are called hyperparameters. Hyperparameter tuning is a commonly used method of model tuning, such as adjusting the number of training times (Epoch) [37], the learning rate [8], the single-step operation volume (Batch Size), etc. The Learning Rate is usually in the range of 0.0 and 1.0, and commonly used ones are 0.01, 0.001 and 0.0001,

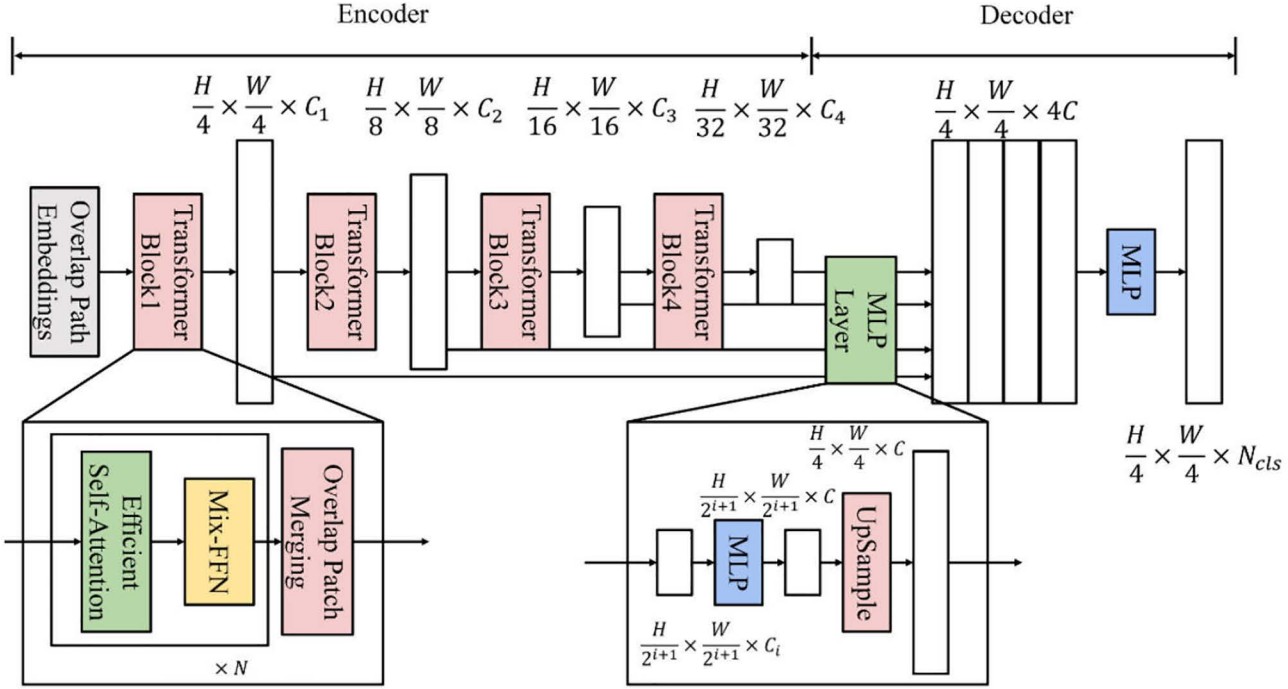

**Fig 2. SegFormer network structure.** https://doi.org/10.6084/m9.figshare.30041161.

etc. A larger Learning Rate can make the model learn more; a smaller Learning Rate is good for the model to learn to a better or even globally optimal. A larger Learning Rate allows the model to learn faster; a smaller Learning Rate is good for the model to learn better or even globally optimal model parameters, but it may take a longer time (Epoch) to train. In extreme cases, too large a Learning Rate will result in too many weight updates and thus be susceptible to the data, causing the model accuracy to oscillate during the training period; while too small a Learning Rate may make training slow and the algorithm difficult to tune. In general, a smaller Learning Rate will require a larger Epoch; moreover, considering the noise estimation of the error gradient, a smaller Batch Size is more suitable for a smaller Learning Rate. Training logs were recorded and visualized using TensorBoard [38] to monitor the learning process.

## Model application

Model application is the process of extracting buildings by applying building extraction models to real-world scenarios. Model application includes model inference, model post-processing, model evaluation and so on.

Model inference refers to using trained models to compute and analyze data to solve real problems. It is usually difficult to train a model with high accuracy and good generalization at one time, and you may need to perform model tuning to improve the model accuracy after the model is trained. In general, model tuning usually starts from three aspects: tuning the training data, tuning the model algorithm, and tuning the training hyperparameters. First, adjusting the quality of the training data to improve the model accuracy can usually be done by using data enhancement techniques to increase the amount of data, supplementing similar samples for poor inference, and eliminating poor quality samples. Second, adjusting the model algorithm involves trying different algorithms during model training to find the best performing algorithm. Different algorithms are suitable for different data and use scenarios, if the training data is small but the model algorithm network is more complex, the model may learn some irrelevant features leading to model overfitting; while the model

algorithm network structure is too simple, it may not be able to fully learn the data features leading to model underfitting, these problems can be adjusted to improve the effect of the model by adjusting the model algorithm. Finally, adjust the training hyperparameters. In deep learning models, the parameters that need to be configured according to the situation before training are called hyperparameters. Hyper-parameter tuning is a commonly used method of model tuning, such as adjusting the number of training times (Epoch), the learning rate, etc. These parameters play an important role in avoiding over (under) fitting, and improving the overall model accuracy, and can be used to find more appropriate parameter values through multiple trials with the help of model training logs.

Post-processing is the data processing of the inference results. The results are optimised by raster data and vector data post-processing tools, raster optimisation mainly includes expansion, contraction, boundary clean-up, plurality filtering, nibbling, region grouping, binary raster refinement; vector data processing includes building regularisation, region aggregation. Building regularization optimizes building contours by extracting key points and the main direction to eliminate irregular boundaries. The steps are as follows:

(1) Key points extraction: Detect corners/edges using algorithms like Harris. $P = \{p_i | p_i = (x_i, y_i)\}$

(2) Main direction calculation: Use PCA to derive the primary orientation θ.

(3) Regularization method selection:

  Right angles: Adjust edges to orthogonal directions.

  Any angles: Smooth edges while preserving angles.

(4) Boundary offset: Constrain adjustments within a buffer of width.

(5) Area filtering: Remove regions with area.

(6) Hole filling: Fill holes smaller than.

(7) Output: Add a status field (0: unprocessed; 1: regularized).

Model Evaluation can be to understand the performance of the model on the test set, quantifying the accuracy of the inference results while also comparing the accuracy of different models. In addition, the model evaluation result usually contains multiple evaluation indexes for the overall and subclasses, which can quickly find out the possible problems during the training process and optimize the model in a targeted way. The model evaluation function takes the vector dataset generated by inference and the corresponding ground truth vector labels as inputs, and generates evaluation indicators in tabular form. The main evaluation indicators include IoU, Precision, F1 score, and Recall.

## Experimental and analysis

### Dataset

The experimental dataset used in this paper is the public dataset WHU building dataset [14] produced by Prof. Shunping Ji's team at Wuhan University. The dataset is publicly available at the official website of the Geospatial Data and Computing Platform (GPCV) group at Wuhan University: http://gpcv.whu.edu.cn/data/building_dataset.html. WHU building dataset is a manually edited dataset of building samples from aerial and satellite images. The aerial dataset consists of more than 220,000 individual buildings extracted from aerial images of Christchurch, New Zealand at a spatial resolution of 0.075 metres and covering an area of 450 square kilometres. The satellite imagery dataset consists of two subsets. One subset is collected from cities around the world and various remote sensing resources, including QuickBird, Worldview series, IKONOS, ZY-3 and others. The other satellite architecture sub-set consists of six adjacent satellite images covering 860 square kilometres of East Asia with a ground resolution of 0.45 metres. This test area is mainly used to evaluate and

develop the generalisation ability of deep learning methods to buildings with different data sources but similar architectural styles in the same geographical area. The vector building map is also fully manually delineated and contains 34,085 buildings. To facilitate training and testing, we seamlessly cropped the training dataset images into 1673 1024 × 1024 tiles.

As shown in Table 1, 25,749 buildings (1,673 tiles) were used for training and the remaining 8,358 buildings were used for testing, for a total of 34,085 buildings used for validation.

## Quantitative evaluation of model building

**Model training hyperparameters.** In deep learning models, the parameters that need to be configured according to the situation before training are called hyperparameters. Hyperparameter tuning is a commonly used method of model tuning, such as adjusting Epoch, Learning Rate, etc., and these parameters play an important role in avoiding over (under) fitting and improving overall model accuracy. The model training hyperparameters are shown in Table 2.

**Model training evaluation metrics.** The x-axis in the training log is usually the Epoch or Iterator counts; the y-axis is often different evaluation metrics to indicate the accuracy of the model, commonly used are Loss [8], Accuracy, IoU and F1 [32]. Model convergence with a small generalisation gap is the goal of training, and the generalisation gap is the difference between the model's performance on the training and validation sets, which is due to the difference in data between the training and validation sets. Underfitting indicates that the model did not learn the features of the training dataset sufficiently, indicating that the training process was stopped too early and that the model's performance could be improved with further training. In the case of the training log, underfitting usually manifests itself in two typical ways. It may show a horizontal curve that levels off or maintains a relatively high loss value and a large difference between the validation and training set metrics, indicating that the model is unable to learn enough information from the training set. This is a good time to consider increasing the complexity of the model by increasing the number of hidden layers to improve the learning ability of the model. Overfitting means that the model learns the training dataset "too" well, or even learns the noise or random error information in the training set. The problem with overfitting is that the model works well on the training set, but generalises poorly to new data. This increase in generalisation error can be measured by the model's performance on the validation set. Using IoU as an example, an overfitting situation can be represented on the

**Table 1. Datasets used in this paper.**

| Dataset | Category | Building Number | Resolution |
|---------|----------|-----------------|------------|
| WHU | train | 25749 | 0.45 m |
| | Val | 34085 | |
| | test | 8358 | 0.45 m |

https://doi.org/10.6084/m9.figshare.30041623

**Table 2. Training configuration of the dataset.**

| Item | WHU |
|------|-----|
| Train size | 1024 × 1024 |
| Epoch | 100 |
| Model | SegFormer |
| Learning rate | 0.001 |
| Backbone network | Mit-b3、Mit-b5 |
| Training log path | E:\log |

https://doi.org/10.6084/m9.figshare.30041638

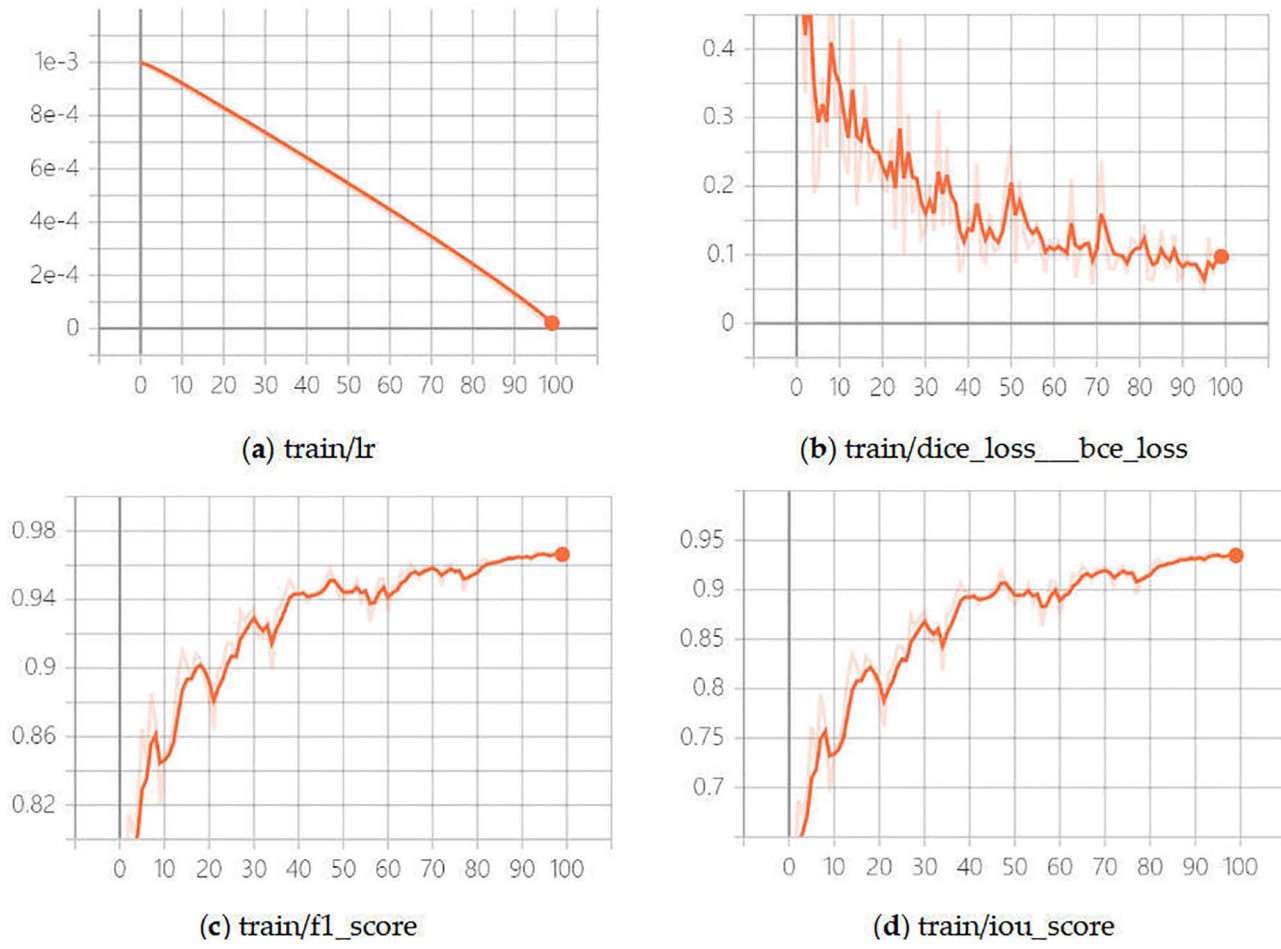

**Fig 3. Training log.** (a) Train/LR; (b) Train/dice_loss_BEC_loss; (c) Ttrain/F1_score; (d) **Train/IoU_score.** https://doi.org/10.6084/m9.figshare.30041170.

training logs as the IoU on the training set continues to increase with experience while the IoU on the validation set rises to a point and begins to decrease again, this inflection point may be the point where training stops as the model is in an overfitted state after that point.

Taking this training as an example, it can be seen that Loss continues to decline, indicators such as F1-Score and IoU rise and gradually smooth out with the increase in the number of training sessions, while the generalisation gap between the training set and the validation set is small. Model convergence is the goal of training, take the evaluation metric of loss as an example, the training log situation of model convergence should be that Loss drops to a stable point on the training set, Loss drops to a stable point on the validation set, and the generalisation gap between the two is small, almost zero in the ideal case. The generalisation gap is the difference in the model's performance on the training and validation sets and is due to the data differences between the training and validation sets. It can be seen that there are some fluctuations and differences in the model's loss on the training and validation sets, and the model has not yet reached optimal convergence. The subsequent experiments can increase the number of experiments to find the inflection point of IoU decline. The training log is shown in Fig 3.

It is usually difficult to train a model with high accuracy and good generalisation in one go, and model tuning may be required after model training to improve model accuracy. Generally speaking, model tuning usually starts from three aspects: adjusting

the training data, adjusting the model algorithm, and adjusting the training hyperparameters. Training data tuning is to improve the data quality from the source so as to improve the model accuracy, which can usually be done by using data enhancement techniques to increase the amount of data, supplementing similar samples for poor inference, and eliminating poor-quality samples. Adjusting the model algorithm is to try different algorithms during model training to find the best performing algorithm. Different algorithms are suitable for different data and use scenarios, if the training data is small but the chosen model algorithm network is more complex, the model may learn some irrelevant features leading to model overfitting; while the model algorithm network structure is too simple, it may not be able to fully learn the data features leading to model underfitting, these problems can be adjusted to improve the effect of the model by adjusting the model algorithm. In deep learning models, the parameters that need to be configured according to the situation before training are called hyperparameters. Hyper-parameter tuning is a commonly used method of model tuning, such as adjusting Epoch, Learning Rate, etc. These parameters play an important role in avoiding overfitting or underfitting, and improving the overall model accuracy, and the model training logs can be used to find more appropriate parameter values through multiple trials.

**Negative lable production.** From the collected dataset images with riverside farmland and motorways but without the presence of buildings are filtered based on the labels and an empty vector labelled dataset is produced, then the data is sliced to generate negative samples [36] using training data generation to address these misidentification cases of riverside farmland and motorways.

## Post processing

Regularise the contour lines of the region of the building coverage area by extracting the key points of the building and the main direction of the building, which is used to eliminate irregular boundaries and details in the geometry of the building range. Set the minimum area of holes inside the region elements. That is, holes smaller than this area will be filled. In Fig 4, we show the inference error data, which is optimized by analysis using hole-filling, and the optimized results are largely consistent with the labels.

Building regularization also includes arbitrary corner regularisation, right angle and diagonal regularisation, direct regularisation and so on. Arbitrary angle regularisation: used for irregular buildings. Right angle and diagonal regularisation: used for buildings defined by right angles and diagonal edges. Right Angle Regularisation: for buildings defined mainly by right angles. In Fig 5, we show the inference data, which was optimized by building right-angle regularization, and the optimized results are largely consistent with the labels.

## Model evaluation

Model evaluation can help us understand the performance of the model on the test set, quantify the accuracy of the inference results and compare the accuracy of different models. In addition, the model evaluation results usually contain multiple evaluation indexes for overall and subclasses, which can quickly identify possible problems during the training process and optimize the model in a targeted way.

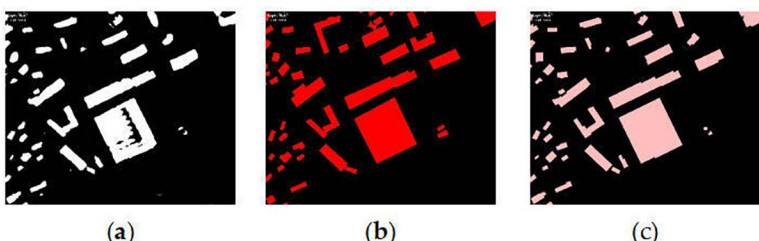

**(a)** **(b)** (c)

**Fig 4. Inference error data.** (a) Inferred results; (b) Validated label; (c) Optimised results. https://doi.org/10.6084/m9.figshare.30041194.

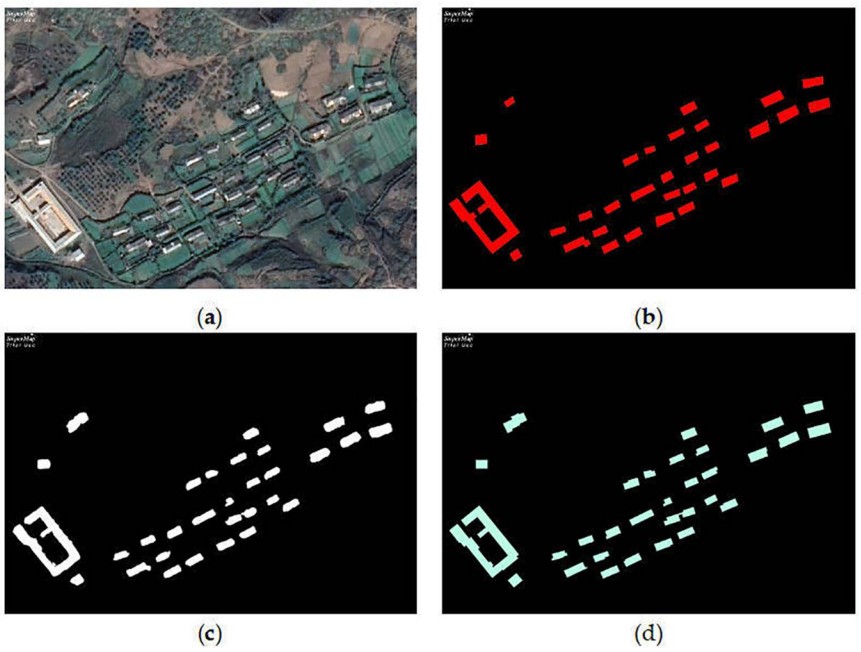

**Fig 5. Building regularization data.** (a) Raw image; (b) Validated label; (c) Inferred results; (d) Optimised results. This schematic diagram is redrawn based on experimental results and is not exactly the same as the original annotated images provided in the WHU dataset. It is only for illustrative purposes. https://doi.org/10.6084/m9.figshare.30041197.

For the model evaluation, the metrics proposed in were used to evaluate the quality of the results obtained in the segmentation task. The following metrics were used: precision (P); recall (R); Intersection-Over-Union (IoU); and F1 score. These are presented in Equations (1)–(4). The symbols in the formulae indicate elements of the confusion matrix, where: TP—True Positive—number of pixels correctly classified as buildings; FP—False Positive—number of background pixels classified as buildings; TN—True Negative—number of pixels correctly classified as background; and FN—False Negative—number of pixels of buildings classified as background. The average IoU value for both classes—buildings and background—was used in the presentation of the results.

$$\text{Precision} = \frac{TP}{TP + FP} \tag{1}$$

$$\text{Recall} = \frac{TP}{TP + FN} \tag{2}$$

$$\text{IoU} = \frac{TP}{TP + FP + FN} \tag{3}$$

$$\text{F1 score} = \frac{2 \times \text{Precision} \times \text{recall}}{\text{Precision} + \text{recall}} = \frac{2 \times TP}{2 \times TP + FP + FN} \tag{4}$$

The input of the model evaluation function is the vector dataset and the real vector labels obtained by inference, and the output is the evaluation table data. The input to the model evaluation function is the vector dataset and real vector labels obtained by inference, and the output is the evaluation table data. We compare the results of two different datasets under

the same model and backbone network, illustrating that different data produce different results. In addition, the MiT-B3 backbone network achieves 65.73% IoU with only 10 trainings, and the MiT-B5 backbone network achieves 94.13% IoU with 100 trainings. These experiments show that with only a small number of trainings, MiT-B3 can complete the inference very quickly, and that MiT-B5 performs better with an increase in the number of trainings. The model evaluation results are shown in Table 3.

### Ablation study

To validate the effectiveness of key components in the proposed method, we conducted an ablation study on the WHU dataset using the SegFormer model with the MiT-B5 backbone. The experiments evaluated the impact of post-processing techniques (e.g., building regularization) and the choice of backbone networks (MiT-B3 vs. MiT-B5) on building extraction performance. Building regularization and other post-processing techniques improved IoU by 4.92% and F1 score by 3.47%, demonstrating their critical role in refining building boundaries and eliminating noise. The MiT-B5 backbone outperformed MiT-B3 by 28.40% in IoU, highlighting the importance of model capacity for complex feature extraction.

## Discussion

### Comparison of different classical methods

In Fig 6, SegFormer outperforms DeepLabV3 on some image segmentation benchmark datasets (e.g., WHU, Cityscapes, and ADE20K). The SegFormer method can be preferred for building extraction.

### Generalizations discussion

The advantages of SegFormer algorithm mainly include: high accuracy: SegFormer algorithm shows high accuracy in image segmentation tasks, and is able to effectively recognize and classify various objects and features in images. High efficiency: Due to the use of techniques such as null convolution, the SegFormer algorithm has a high computational efficiency when dealing with large-scale image data, and is able to quickly complete the image segmentation task. Flexibility: SegFormer algorithm has high flexibility and can be adjusted and optimized according to different tasks and datasets to obtain better segmentation results. However, SegFormer algorithm also has some drawbacks: long training time: since SegFormer algorithm adopts deep neural network, it needs a lot of computational resources and time in the training process.

High model complexity: the SegFormer algorithm has a high model complexity and requires a large number of parameters and computations, which makes the model less interpretable and portable. Poor segmentation of small targets: the

**Table 3. Model evaluation results.**

| Methods | BackBone | Data | IoU | Precision | F1 | Recall |
|---|---|---|---|---|---|---|
| FCN [5] | ResNet-101 | ADE20K | 41.4 | – | – | – |
| OCRNet [5] | HRNet-W48 | ADE20K | 45.6 | – | – | – |
| SegFormer [5] | Mit-B3(S1-4) | ADE20K | 48.6 | – | – | – |
| SETR [5] | ViT-Large | ADE20K | 50.2 | – | – | – |
| SegFormer(Ours) | Mit-B3 | WHU | 65.73 | 80.12 | 79.32 | – |
| SegFormer [7] | Transconv | WHU | 74.23 | 87.35 | 84.27 | 83.61 |
| DeeplabV3+ [39] | – | WHU | 89.19 | 94.95 | 94.29 | 93.64 |
| U-Net++ [39] | – | WHU | 89.36 | 95.34 | 94.38 | 93.44 |
| SegFormer(Ours) | Mit-B5 | WHU | 94.13 | 96.70 | 96.58 | – |

https://doi.org/10.6084/m9.figshare.30041641

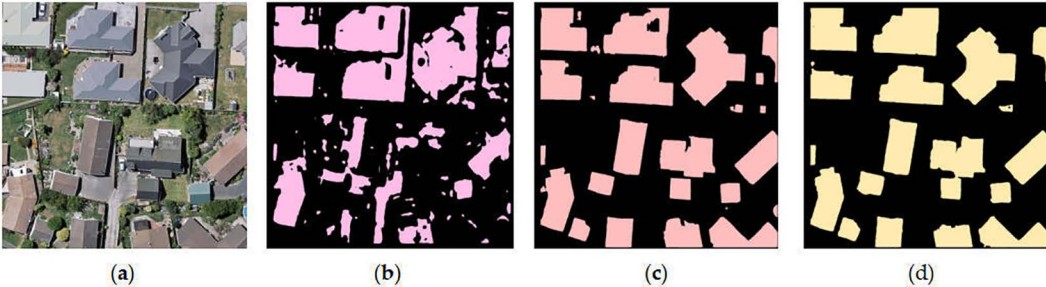

**Fig 6. Comparison of Different Classical Methods.** (a) Raw image; (b) DeeplabV3+; (c) SegFormer(Mit-B3); (d) SegFormer(Mit-B5). This schematic diagram is redrawn based on experimental results and is not exactly the same as the original annotated images provided in the WHU dataset. It is only for illustrative purposes. https://doi.org/10.6084/m9.figshare.30041209.

segmentation effect of the SegFormer algorithm may suffer when dealing with small targets, because the features of small targets are difficult to be captured and recognized by the model. A comparative analysis of IOUs in building extraction experiments is shown in Fig 7.

## Limitations and future direction

The experimental results show that the intelligent binary classification limitations of remote sensing images mainly come from different data volume, different data quality, feature diversity, and model generalisation ability. Data volume limitation: intelligent binary classification methods usually rely on a large amount of remote sensing image data for training. However, in some cases, it may be challenging to obtain a large amount of high-quality remote sensing image data, which may lead to degradation of model performance. Data quality limitations: the quality of remotely sensed images may be affected by various factors, such as weather conditions, lighting conditions, sensor performance, etc. These factors may lead to degradation of image quality, which may affect the accuracy of classification results. Feature diversity: Remote sensing images contain a rich variety of feature types, such as buildings, roads, water bodies, vegetation, and so on. However, intelligent binary classification methods usually can only handle two types of features, which cannot fully explore and utilise the rich information in remote sensing images. Model generalisation ability: intelligent binary classification methods

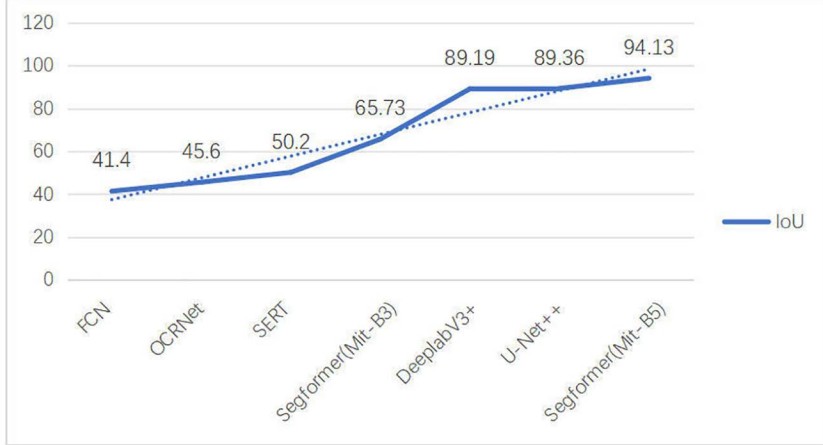

**Fig 7. Analysis of building extraction experiments.** https://doi.org/10.6084/m9.figshare.30041215.

may have a strong dependence on training data, and when faced with new remote sensing image data, the generalisation ability of the model may be limited.

Considering these limitations, several future directions can be pursued to enhance the performance and applicability of intelligent binary classification methods for remote sensing images. These include data augmentation techniques to expand the dataset, fusion of multi-source data to incorporate various types of remote sensing data, reinforcement learning to optimize model performance through interactive learning, improving model interpretability for better understanding of the decision-making process, and fostering cross-domain collaboration to promote the development of remote sensing image intelligence classification technology.

## Conclusions

The proposed SegFormer-based building extraction framework demonstrates consistent performance across urban, rural, and mountainous environments. Quantitative evaluation on the WHU dataset reveals the MiT-B5 variant achieves superior 94.13% IoU accuracy, while the MiT-B3 configuration offers faster processing speeds at 65.73% IoU, providing practical options for different application scenarios. Cross-region validation confirms the method's robustness when trained with geographically diverse samples.

Current limitations involve occasional misclassification of non-building features and omissions in complex urban layouts. Future research will pursue two key directions: first, developing foundation models requiring minimal or no pre-training for remote sensing interpretation tasks; second, enhancing negative sampling techniques and domain-specific optimization for challenging scenarios like shadowed areas and dense building clusters. Additional improvements will incorporate temporal analysis to address seasonal variations in image quality.

This work establishes a reliable foundation for large-scale building extraction with direct applications in urban development monitoring and emergency response systems. The accompanying release of implementation code and trained models supports reproducibility and community-driven advancement in geospatial analysis. The framework's adaptability positions it as a valuable tool for next-generation geographic information systems and smart city infrastructure planning.

## Supporting information

**S1 Fig. A research framework for building extraction from remote sensing images.**
(TIF)

**S2 Fig. SegFormer network structure.**
(TIF)

**S3 Fig. Training log. (a) Train/LR; (b) Train/dice_loss_BCE_loss; (c) Train/F1_score; (d) Train/IoU_score.**
(TIF)

**S4 Fig. Inference error data. (a) Inferred results; (b) Validated label; (c) Optimised results.**
(TIF)

**S5 Fig. Building regularization data. (a) Raw image; (b) Validated label; (c) Inferred results; (d) Optimised results.**
(TIF)

**S6 Fig. Comparison of different classical methods. (a) Raw image; (b) DeeplabV3 + ; (c) SegFormer(Mit-B3); (d) SegFormer(Mit-B5).**
(TIF)

**S7 Fig. Analysis of building extraction experiments.**
(TIF)

**S1 Table. Datasets used in this paper.**
(DOC)

**S2 Table. Training configuration of the dataset.**
(DOC)

**S3 Table. Model evaluation results.**
(DOC)

## Acknowledgments

The authors would like to thank the team from Wuhan University for providing the WHU building dataset, which was used for the initial model training and validation in this study. For the final figures presented in this manuscript, open-source satellite imagery from the Landsat 8 was used to generate model inference and post-processing results, ensuring compliance with the CC BY 4.0 license.

## Author contributions

**Conceptualization:** Deliang Li, Tao Liu.

**Data curation:** Deliang Li, Haokun Wang.

**Formal analysis:** Deliang Li.

**Funding acquisition:** Deliang Li, Haokun Wang.

**Investigation:** Long Yan.

**Methodology:** Deliang Li, Haokun Wang.

**Resources:** Deliang Li, Haokun Wang.

**Software:** Deliang Li, Haokun Wang.

**Validation:** Deliang Li, Haokun Wang.

**Writing – original draft:** Deliang Li, Long Yan.

**Writing – review & editing:** Deliang Li, Long Yan.

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
