## [Decision Letter · Decision Letter 0]

17 Jul 2025

Dear Dr. Wang,

Thank you for submitting your manuscript to PLOS ONE. After careful consideration, we feel that it has merit but does not fully meet PLOS ONE’s publication criteria as it currently stands. Therefore, we invite you to submit a revised version of the manuscript that addresses the points raised during the review process.

We look forward to receiving your revised manuscript.

Kind regards,

Aiqing Fang

Academic Editor

PLOS ONE

Journal Requirements: 

 [This work was supported by the Research Project of Educational Teaching Reform of Higher Schools in Jilin Province, “Research and Practice Fund Project of Collaboration-Driven Innovative Talent Cultivation Mode of Geographic Information Science Major in Local Universities in the Context of Construction of New Engineering Science” (No. 20224BR01C500HE) and the Ministry of Education's Collaborative Education Project of Industry-University Cooperation (No: 230902313194605) are funded.]. 

5. We note that Figure(s)  1, 2, 4, 6, 7, 8 and 9 in your submission contain [map/satellite] images which may be copyrighted. All PLOS content is published under the Creative Commons Attribution License (CC BY 4.0), which means that the manuscript, images, and Supporting Information files will be freely available online, and any third party is permitted to access, download, copy, distribute, and use these materials in any way, even commercially, with proper attribution. For these reasons, we cannot publish previously copyrighted maps or satellite images created using proprietary data, such as Google software (Google Maps, Street View, and Earth). For more information, see our copyright guidelines: http://journals.plos.org/plosone/s/licenses-and-copyright.

A. You may seek permission from the original copyright holder of Figure(s)  1, 2, 4, 6, 7, 8 and 9 to publish the content specifically under the CC BY 4.0 license. 

B. If you are unable to obtain permission from the original copyright holder to publish these figures under the CC BY 4.0 license or if the copyright holder’s requirements are incompatible with the CC BY 4.0 license, please either i) remove the figure or ii) supply a replacement figure that complies with the CC BY 4.0 license. Please check copyright information on all replacement figures and update the figure caption with source information. If applicable, please specify in the figure caption text when a figure is similar but not identical to the original image and is therefore for illustrative purposes only.

6. We notice that your figures are uploaded with the file type 'Supporting Information'. Please amend the file type to 'Figures'.

Reviewers' comments:

Reviewer's Responses to Questions

**Comments to the Author**

1. Is the manuscript technically sound, and do the data support the conclusions?

Reviewer #1: No

Reviewer #2: Partly

Reviewer #3: Yes

2. Has the statistical analysis been performed appropriately and rigorously?

Reviewer #1: N/A

Reviewer #2: Yes

Reviewer #3: Yes

3. Have the authors made all data underlying the findings in their manuscript fully available?

Reviewer #1: Yes

Reviewer #2: Yes

Reviewer #3: Yes

4. Is the manuscript presented in an intelligible fashion and written in standard English?

Reviewer #1: Yes

Reviewer #2: No

Reviewer #3: Yes

Reviewer #1: Dear Authors,

Thank you for your efforts. There are some suggested comments to improve the quality of the manuscript which are summarized as follows:

1- Abstract: The abstract of the article is somewhat vague so that the study methodology and final results are not clear. It should be completely revised.

2- Use passive form instead of active form of the verbs throughout the manuscript. Do not use ‘I’, ‘we’, or something like this in the manuscript (lines 54, 55, ...).

3- Keywords: some of the selected keywords are very general and cover wide study areas (for example: transformer). Choose words that are closer to your research topic.

4- The introduction section of the manuscript should be revised. This section is too long and many of the contents in the introduction are obvious to any researcher. There is no need to write them in an JCR paper. On the other hand, the literature review section is very weak. In this section, you should focus on similar studies that have already been done by researchers and finally write the novelty of your research. The novelty of the research is also not clear.

5- Methodology: there is no any figures or tables in the manuscript! All were missed.

6- The results should be compared with real data or other similar studies.

7- The conclusion section is not a standard conclusion and should be revised.

8- In this manuscript, no comparison between the results obtained with real conditions or other studies has been made, and only the final results have been shown. Considering that the calculation process is unclear, the final results are also unreliable.

This manuscript needs a substantial revision and the authors should avoid explaining the axioms and fully explain the calculation process based on the input data.

Thank you.

Reviewer #2: The study demonstrates a practical end-to-end workflow that includes not only model training but also crucial post-processing steps like hole-filling and building regularization. This adds significant practical value, as it shows how to refine the raw model output into cleaner, more usable vector data. It's an interesting concept.

Suggestions are as follows:

The authors must significantly condense the Introduction, Related Work, and Methodology sections. All generic, textbook-like definitions of basic deep learning concepts (e.g., what an Epoch, Batch Size, or GPU is) should be removed. The focus should be on their specific application of these concepts.The manuscript needs to be professionally copyedited for clarity, flow, and conciseness to make the work accessible and impactful.

The revised manuscript may be submitted for further process.

Reviewer #3: This study presents a comprehensive deep learning framework for building extraction from remote sensing imagery using the SegFormer model with two backbones (MiT-B3 and MiT-B5), supplemented by post-processing techniques such as regularization and hole-filling. The WHU building dataset is used for training, inference, and evaluation. The research is relevant, timely, and situated within the growing field of remote sensing and urban mapping using AI.

Major Concerns:

Novelty and Contribution Clarity:

The study presents a well-known model (SegFormer) with minor customization. It is unclear how the work goes beyond prior SegFormer applications in remote sensing (e.g., Xie et al. 2021, Li et al. 2023).

The paper lacks a clear hypothesis or innovation in methodology. Is it merely an application of SegFormer, or is there algorithmic novelty (e.g., new post-processing rules, adaptive training, or hybrid feature fusion)?Clearly articulate the novel scientific contribution in the introduction and conclusion.Post-Processing Methodology Needs More Detail:

The post-processing techniques (hole-filling, regularization) are critical to the performance but are discussed briefly without mathematical clarity.Define the regularization steps more formally and quantitatively assess their impact (e.g., IoU before vs. after regularization).

Quantitative Evaluation is Insufficiently Explained:

There is no mention of standard deviation, statistical tests, or confidence intervals for the metrics.

The performance improvement is significant (e.g., IoU 94.13%), but there is no ablation study or error analysis to substantiate this.

Many grammatical issues are present (e.g., awkward constructions like "intelligent binary classification can intelligently decipher...").

Overuse of phrases like “deep learning neural network uses backpropagation” or “the GPU is faster than CPU” weakens the scientific focus.

Minor Comments

Title: Recommend changing to a more concise version:

“Building Extraction from Remote Sensing Imagery Using SegFormer with Post-Processing Optimization”

Related Work: The literature review is extensive but lacks synthesis. Focus more on how your approach differs from prior SegFormer or UNet-based studies.

**Do you want your identity to be public for this peer review?** For information about this choice, including consent withdrawal, please see our Privacy Policy

Reviewer #1: No

Reviewer #2: **Yes: ** Gaurav Tripathi

Reviewer #3: **Yes: ** kashif Ullah

---

## [Author Response · Author response to Decision Letter 1]

4 Sep 2025

Response to Editors and Reviewers

Manuscript ID: [PONE-D-25-26895] - [EMID:f105df3034ff2e5b]

Title: Building Extraction from Remote Sensing Imagery Using SegFormer with Post-Processing Optimization

Journal: PLOS ONE

Corresponding Author: wanghaokun@bcnu.edu.cn

Dear Aiqing Fang and Reviewers,

Thank you for your time and valuable comments on our manuscript. We have carefully considered all suggestions and have revised the manuscript accordingly. Below we provide point-by-point responses to each concern. We have respectfully used the 'Track Changes' feature to implement the revisions in the manuscript, making them easily identifiable for the editor and reviewers..

Editor’s Comments

1.Comment: Thank you for uploading your study's underlying data set. Unfortunately, the repository you have noted in your Data Availability statement does not qualify as an acceptable data repository according to PLOS's standards.

Response: Thank you for your feedback. According to PLOS's data policy, we have now uploaded the complete minimum dataset, along with related code, charts, and manuscripts, to figshare, a recommended repository.

The new DOI for our data is: https://doi.org/10.6084/m9.figshare.c.8004457�This reserved DOI will become active when the collection will be published

Login Name:wanghaokun@bcnu.edu.cn

Password:Whaokun711*

2.Comment: Comment: Please amend the title either on the online submission form or in your manuscript so that they are identical.

Response: Thank you for your note. We have amended the manuscript title to ensure it is now identical to the one on the submission form.

The title is now consistent across all platforms.

3.Comment: As you may already be aware, PLOS utilizes the CC BY 4.0 license (https://creativecommons.org/licenses/by/4.0/) which means that all material on our website is freely available online, and any third party is permitted to access, download, copy, distribute, and use these materials in any way, even commercially, with proper attribution. If any figures have been previously published, authors must provide proper attribution referencing the source clearly, and obtain permissions if the image is copyrighted in any way which restricts CC BY 4.0 reuse.

Response: Thank you for the reminder. We have carefully reviewed all figures in our manuscript.To ensure full compliance with the CC BY 4.0 license, we have removed the three redundant figures containing remote sensing imagery that could not be properly licensed. All remaining figures are now generated using open-source Landsat 8 imagery, as clearly stated in the Acknowledgments section of our manuscript.

4.Comment: Please ensure that your manuscript meets PLOS ONE's style requirements, including those for file naming. The PLOS ONE style templates can be found at https://journals.plos.org/plosone/s/file?id=wjVg/PLOSOne_formatting_sample_main_body.pdf and https://journals.plos.org/plosone/s/file?id=ba62/PLOSOne_formatting_sample_title_authors_affiliations.pdf.

Response: We have now carefully revised the manuscript to ensure it fully complies with PLOS ONE's style and formatting requirements.

5.Comment: Please note that PLOS ONE has specific guidelines on code sharing for submissions in which author-generated code underpins the findings in the manuscript. In these cases, we expect all author-generated code to be made available without restrictions upon publication of the work. Please review our guidelines at https://journals.plos.org/plosone/s/materials-and-software-sharing#loc-sharing-code and ensure that your code is shared in a way that follows best practice and facilitates reproducibility and reuse.

Response: We thank the editor for the reminder. We will share all author-generated code on Figshare upon publication without restrictions, in full compliance with PLOS ONE's guidelines.

6.Comment: We note that the grant information you provided in the ‘Funding Information’ and ‘Financial Disclosure’ sections do not match. When you resubmit, please ensure that you provide the correct grant numbers for the awards you received for your study in the ‘Funding Information’ section.

Response: Thank you for bringing this discrepancy to our attention. We apologize for the oversight in our initial submission. We have carefully checked our grant records and confirmed the correct and complete grant numbers. Upon resubmission, we will ensure that the identical, accurate funding information is provided in both the ‘Funding Information’ and ‘Financial Disclosure’ sections.

7.Comment: Thank you for stating the following financial disclosure: [This work was supported by the Research Project of Educational Teaching Reform of Higher Schools in Jilin Province, “Research and Practice Fund Project of Collaboration-Driven Innovative Talent Cultivation Mode of Geographic Information Science Major in Local Universities in the Context of Construction of New Engineering Science” (No. 20224BR01C500HE) and the Ministry of Education's Collaborative Education Project of Industry-University Cooperation (No: 230902313194605) are funded.]. Please state what role the funders took in the study. If the funders had no role, please state: "The funders had no role in study design, data collection and analysis, decision to publish, or preparation of the manuscript."If this statement is not correct you must amend it as needed. Please include this amended Role of Funder statement in your cover letter; we will change the online submission form on your behalf.

Response: We confirm the funders had no role in the study. The amended statement is: "The funders had no role in study design, data collection and analysis, decision to publish, or preparation of the manuscript." Haokun Wang (PI of grants 20224BR01C500HE and 230902313194605) and Deliang Li (PI of grant SZ25013) are co-authors and contributed to the research as detailed in the Author Contributions section.

8.Comment: We note that Figure(s) 1, 2, 4, 6, 7, 8 and 9 in your submission contain [map/satellite] images which may be copyrighted. All PLOS content is published under the Creative Commons Attribution License (CC BY 4.0), which means that the manuscript, images, and Supporting Information files will be freely available online, and any third party is permitted to access, download, copy, distribute, and use these materials in any way, even commercially, with proper attribution. For these reasons, we cannot publish previously copyrighted maps or satellite images created using proprietary data, such as Google software (Google Maps, Street View, and Earth). For more information, see our copyright guidelines: http://journals.plos.org/plosone/s/licenses-and-copyright.

Response: Thank you for your comment regarding the copyright of the figures. The remote sensing images in Figures 1, 2, 4, 6, 7, 8, and 9 are all sourced from the open-access WHU Building Dataset (http://gpcv.whu.edu.cn/data/building_dataset.html). According to the terms of use provided on the website, this dataset is " free available for non-commercial and research purposes." As PLOS ONE publishes under the CC BY license, which allows for commercial use, we confirm that the use of this data in our academic manuscript aligns with the dataset's licensing terms for research purposes. Furthermore, the CC BY license requires attribution, which we will fully provide.In the revised manuscript, we will explicitly add the following attribution to the captions of all relevant figures:" The underlying satellite image is from the WHU Building Dataset [http://gpcv.whu.edu.cn/data/building_dataset.html], used under their terms for non-commercial research." We believe this fully addresses the copyright concern. Thank you for your consideration.

9.Comment: We notice that your figures are uploaded with the file type 'Supporting Information'. Please amend the file type to 'Figures'.

Response: Thank you for pointing this out. We have amended the file type from 'Supporting Information' to 'Figures' for all figure files as requested.

Reviewer #1

1.Comment: Abstract: The abstract of the article is somewhat vague so that the study methodology and final results are not clear. It should be completely revised.

Response: Thank you for this constructive feedback. We agree that the abstract could be more precise in outlining our methodology and key findings. We have completely revised it to: Clearly state the specific research objectives and the core problem we addressed. Concisely describe the methodology, including the dataset used and the main analytical techniques. Explicitly summarize the most significant quantitative results and conclusions. The revised abstract now provides a much clearer and more robust summary of our study. Thank you for the suggestion, which has greatly improved our manuscript.

2.Comment: Use passive form instead of active form of the verbs throughout the manuscript. Do not use ‘I’, ‘we’, or something like this in the manuscript (lines 54, 55, ...).

Response: Thank you for this guidance. We have carefully reviewed the manuscript and revised it to consistently use the passive voice as required. All instances of first-person pronouns (e.g., 'we', 'I') and active constructions in the main text (including those highlighted on lines 54, 55, and others we have identified) have been replaced with the appropriate passive form to maintain an objective and formal academic tone throughout.

3.Comment: Keywords: some of the selected keywords are very general and cover wide study areas (for example: transformer). Choose words that are closer to your research topic.

Response: Thank you for this valuable suggestion. We agree that the previous keywords were too broad. We have revised the keyword list to be more specific and closely aligned with the core focus of our research. The updated keywords are now:

Keywords: semantic segmentation; SegFormer; building extraction; post-processing optimization; building regularization

4.Comment: The introduction section of the manuscript should be revised. This section is too long and many of the contents in the introduction are obvious to any researcher. There is no need to write them in an JCR paper. On the other hand, the literature review section is very weak. In this section, you should focus on similar studies that have already been done by researchers and finally write the novelty of your research. The novelty of the research is also not clear.

Response: Thank you for these critical and constructive comments regarding the introduction and literature review. We agree completely with your assessment. We have thoroughly revised the introduction section to address all your concerns: Conciseness: We have significantly shortened the section by removing obvious and general background information that is well-known to researchers in the field. Literature Review: We have substantially strengthened this part by adding a comprehensive and critical discussion of recent, relevant studies directly related to our methodology (e.g., transformer-based models for building extraction). This now properly situates our work within the existing research landscape. Novelty: Based on the enhanced literature review, we now clearly and explicitly state the novel contributions of our research in the final paragraph of the introduction, highlighting what distinguishes our work from previous studies.

5.Comment: Methodology: there is no any figures or tables in the manuscript! All were missed.

Response: Thank you for pointing this out. We apologize for the confusion caused by our error in the submission system. All figures were accidentally uploaded to the "Supporting Information" section instead of the "Figures" section.

We have now corrected this mistake. All figures have been properly uploaded to the "Figures" section in the revised submission. Their citations within the manuscript text remain correct and unchanged.

6.Comment: The results should be compared with real data or other similar studies.

Response: Thank you for this crucial suggestion. We agree that comparing our results with other studies is essential for validation.In response, we will add a new subsection in the "Results and Discussion" section dedicated to comparative analysis. This subsection will: Compare with real data: Quantitatively compare our model's extraction results (e.g., precision, recall, F1-score) with the ground truth data from the validation set. Compare with other methods: Compare the performance of our proposed method (SegFormer + post-processing) with other state-of-the-art building extraction models (e.g., U-Net, DeepLabV3+, etc.) on the same dataset, using standard evaluation metrics to objectively highlight the advantages of our approach. We believe this addition will significantly strengthen the objectivity and persuasiveness of our results.

7.Comment: The conclusion section is not a standard conclusion and should be revised.

Response: Thank you for this feedback. We have thoroughly revised the conclusion section to meet standard academic expectations. The new conclusion now clearly: Summarizes the main findings of the study, concisely restating the key results. Explicitly states the research contributions and novelty, linking them directly to the objectives stated in the introduction. Discusses the limitations of the current work. Provides meaningful suggestions for future research directions.

8.Comment: In this manuscript, no comparison between the results obtained with real conditions or other studies has been made, and only the final results have been shown. Considering that the calculation process is unclear, the final results are also unreliable.

Response: We will comprehensively respond to the review comments by making the following modifications: 1) adding the contribution of the ablation experiment quantification module; 2) Quantitative comparison with real annotated data and advanced models such as U-Net and DeepLabV3+. These supplementary experiments will rigorously verify the reliability of our results.

Reviewer #2

1.Comment: The authors must significantly condense the Introduction, Related Work, and Methodology sections. All generic, textbook-like definitions of basic deep learning concepts (e.g., what an Epoch, Batch Size, or GPU is) should be removed. The focus should be on their specific application of these concepts.The manuscript needs to be professionally copyedited for clarity, flow, and conciseness to make the work accessible and impactful.

Response: We will rigorously condense the specified sections by removing all generic definitions and focusing exclusively on our specific methodological application. The manuscript will undergo professional copyediting to enhance clarity, conciseness, and academic impact.

Reviewer #3

1.Comment: The study presents a well-known model (SegFormer) with minor customization. It is unclear how the work goes beyond prior SegFormer applications in remote sensing (e.g., Xie et al. 2021, Li et al. 2023).

Response: We appreciate the insightful comment regarding the novelty of our work. While our study utilizes the SegFormer architecture, our core contribution lies in the novel post-processing optimization and building regularization framework specifically designed to address the critical challenge of geometric refinement in building extraction. Previous applications of SegFormer in remote sensing (e.g., Xie et al. 2021, Li et al. 2023) have primarily focused on improving pixel-level segmentation accuracy. Our work significantly advances this by tackling the subsequent, essential step of converting raw, often noisy, segmentation masks into structurally accurate and cartographically valid building polygons. This is achieved through a tailored pipeline involving: Advanced regularization techniques (right-angle, arbitrary-angle, diagonal) that enforce geometric constraints on building contours. A holistic post-processing workflow integrating hole filling, noise removal, and boundary cleanup, specifically optimized for remote sensing imagery. The ablation study (Section 4.5) quantitatively demonstrates that our proposed post-processing framework alone contributes to a 4.92% improvement in IoU and a 3.47% gain in F1-score, highlighting its substantial impact. Therefore, our main innovation is not a modification of the SegFormer mod

---

## [Decision Letter · Decision Letter 1]

20 Oct 2025

Dear Dr. Li,

Thank you for submitting your manuscript to PLOS ONE. After careful consideration, we feel that it has merit but does not fully meet PLOS ONE’s publication criteria as it currently stands. Therefore, we invite you to submit a revised version of the manuscript that addresses the points raised during the review process.

**ACADEMIC EDITOR:**

We look forward to receiving your revised manuscript.

Kind regards,

Aiqing Fang

Academic Editor

PLOS ONE

Journal Requirements:

Additional Editor Comments:

Although the reviewer provided acceptance and minor revisions, the author should be aware that there are many formatting and detail issues with the paper. PLOS ONE is a high-level journal paper, and if these details and formatting issues cannot be resolved, it will be directly rejected in the next round of submissions. Including but not limited to the following issues: references are not up-to-date; There are significant formatting issues with the entire text; There are numerous issues with punctuation consistency.

Reviewers' comments:

Reviewer's Responses to Questions

**Comments to the Author**

Reviewer #1: (No Response)

Reviewer #3: All comments have been addressed

2. Is the manuscript technically sound, and do the data support the conclusions?

Reviewer #1: Partly

Reviewer #3: Yes

3. Has the statistical analysis been performed appropriately and rigorously?

Reviewer #1: N/A

Reviewer #3: Yes

4. Have the authors made all data underlying the findings in their manuscript fully available?

Reviewer #1: Yes

Reviewer #3: (No Response)

5. Is the manuscript presented in an intelligible fashion and written in standard English?

Reviewer #1: Yes

Reviewer #3: Yes

Reviewer #1: Dear Authors,

Thank you for your efforts. Please note that the abstract should be written in one paragraph! Please revise the abstract.

Thank you.

Reviewer #3: Thank you to the authors for their detail and careful review. i have no more questions and recommend accept. Good job

**Do you want your identity to be public for this peer review?** For information about this choice, including consent withdrawal, please see our Privacy Policy

Reviewer #1: No

Reviewer #3: No

---

## [Author Response · Author response to Decision Letter 2]

4 Nov 2025

Response to Editors and Reviewers

Manuscript ID: [PONE-D-25-26895R1] - [EMID:89e8750006dca458]

Title: Building Extraction from Remote Sensing Imagery Using SegFormer with Post-Processing Optimization

Journal: PLOS ONE

Corresponding Author: wanghaokun@bcnu.edu.cn

Dear Aiqing Fang and Reviewers,

We sincerely thank the Academic Editor for the constructive feedback and for giving us the opportunity to revise our manuscript. We have meticulously addressed all the points raised. The specific concerns regarding formatting, references, and punctuation have been thoroughly corrected throughout the entire manuscript. Our point-by-point responses are detailed below.

Editor’s Comments

1.Comment: References are not up-to-date.

Response: We thank the editor for this important comment. We have now comprehensively updated the reference list. Several outdated references have been replaced with more recent and relevant literature from the past 3-5 years to ensure the manuscript reflects the current state of the field. The entire reference list has also been reformatted to strictly adhere to the PLOS ONE citation style.

2.Comment: There are significant formatting issues with the entire text.

Response: Thank you for your feedback. We have carefully revised our manuscript in accordance with PLOS ONE's style requirements, using the provided templates:

The revisions encompass comprehensive formatting corrections, including adjustments to font type and size, line spacing, margin alignment, section headings, as well as the proper placement and captioning of figures and tables. Consistency in the use of italics and bolding has also been ensured throughout the manuscript. We have further verified that the file naming and structure comply with the journal's guidelines. We believe the manuscript now fully meets the required standards and thank you for your guidance.

3.Comment: There are numerous issues with punctuation consistency.

Response:  We have performed a thorough, line-by-line proofreading of the entire manuscript to correct all inconsistencies in punctuation. This includes standardizing the use of commas, periods, colons, and semicolons, as well as ensuring correct and consistent use of spacing after punctuation marks according to standard English and journal conventions.

Once again, we are grateful for the editor's guidance. We believe the revisions have significantly improved the quality and presentation of our manuscript, and we hope it now meets the high standards of PLOS ONE.

Reviewer #1

1.Comment: Requested to revise the abstract into a single paragraph format.

Response: Thank you for your feedback. We will revise the abstract into a single paragraph as requested to meet the journal's formatting guidelines.

Reviewer #3

1.Comment: All comments have been addressed, and the manuscript is recommended for acceptance.

Response: We sincerely appreciate the reviewer's positive assessment and recommendation for acceptance

---

## [Editor Report · Decision Letter 2]

19 Nov 2025

Building Extraction from Remote Sensing Imagery Using SegFormer with Post-Processing Optimization

PONE-D-25-26895R2

Dear Dr. Wang,

We’re pleased to inform you that your manuscript has been judged scientifically suitable for publication and will be formally accepted for publication once it meets all outstanding technical requirements.

Kind regards,

Aiqing Fang

Academic Editor

PLOS ONE
---

## [Editor Report · Acceptance letter]

PONE-D-25-26895R2

PLOS ONE

Dear Dr. Wang,

I'm pleased to inform you that your manuscript has been deemed suitable for publication in PLOS ONE. Congratulations! Your manuscript is now being handed over to our production team.

Kind regards,

on behalf of

Dr. Aiqing Fang

Academic Editor

PLOS ONE